# Chronic Hepatitis B Treatment Strategies Using Polymerase Inhibitor-Based Combination Therapy

**DOI:** 10.3390/v13091691

**Published:** 2021-08-26

**Authors:** Eriko Ohsaki, Yadarat Suwanmanee, Keiji Ueda

**Affiliations:** Division of Virology, Department of Microbiology and Immunology, Osaka University Graduate School of Medicine, Suita 565-0871, Japan; ysuwanmanee@virus.med.osaka-u.ac.jp (Y.S.); kueda@virus.med.osaka-u.ac.jp (K.U.)

**Keywords:** hepatitis B virus, polymerase, combination therapy, NRTI, NNRTI, cccDNA

## Abstract

Viral polymerase is an essential enzyme for the amplification of the viral genome and is one of the major targets of antiviral therapies. However, a serious concern to be solved in hepatitis B virus (HBV) infection is the difficulty of eliminating covalently closed circular (ccc) DNA. More recently, therapeutic strategies targeting various stages of the HBV lifecycle have been attempted. Although cccDNA-targeted therapies are attractive, there are still many problems to be overcome, and the development of novel polymerase inhibitors remains an important issue. Interferons and nucleos(t)ide reverse transcriptase inhibitors (NRTIs) are the only therapeutic options currently available for HBV infection. Many studies have reported that the combination of interferons and NRTI causes the loss of hepatitis B surface antigen (HBsAg), which is suggestive of seroconversion. Although NRTIs do not directly target cccDNA, they can strongly reduce the serum viral DNA load and could suppress the recycling step of cccDNA formation, improve liver fibrosis/cirrhosis, and reduce the risk of hepatocellular carcinoma. Here, we review recent studies on combination therapies using polymerase inhibitors and discuss the future directions of therapeutic strategies for HBV infection.

## 1. Introduction

As one of the main causes of liver disease, hepatitis B virus (HBV) infection is a global health problem affecting approximately 296 million individuals worldwide (https://www.who.int/news-room/fact-sheets/detail/hepatitis-b, accessed on 27 July 2021). Chronic hepatitis B (CHB) virus infection can be classified into several phases with variable levels of serum alanine aminotransferase (ALT) activity, which is a marker of liver damage, HBV antigens, including HBsAg, HBeAg, and core antigen, and HBV DNA [1]. All patients with chronic HBV infection are at high risk of developing liver fibrosis and hepatocellular carcinoma (HCC). To increase the survival rate and decrease the risk of disease progression, many research teams have been working on novel strategies to eliminate the virus.

Unfortunately, current therapies against HBV are limited to interferons and NRTIs (nucleos(t)ide reverse transcriptase inhibitors). Treatment by interferon (IFN)-α, which is an immune modulator, was the first approach developed for treatment of CHB. IFN-α was approved in 1991, and it has been replaced by its pegylated form (Peg-IFN-α), which has a significantly extended half-life and more sustained virologic response. Nucleos(t)ide analogues (NAs), also called NRTIs, are potent polymerase inhibitors directly targeting the viral polymerase elongation process via incorporation into replicated DNA, and namely function as chain terminators. Lamivudine, commonly called 3TC, was utilized at first, but because of the high rate of HBV polymerase gene mutants or variants capable of evading its activity, other NRTIs such as entecavir (ETV) and tenofovir have been introduced as further anti-HBV treatments [2,3].

Even if the replication of the viral genome is inhibited by NRTIs, the effect on HBV replication is only transient; once treatment is discontinued, mRNA expression is resumed because of the existence of cccDNA, which leads to the resumption of viral replication [4,5]. This phenomenon has led to the definition of two types of HBV “cure”: the “functional” cure and the “complete” or “virologic” cure [6]. “Functional” cure refers to the persistent disappearance of HBsAg and acquisition of anti-HBs antibodies, as well as normalization of liver enzymes after treatment. “Complete” or “virological” cure adds to these effects the loss of cccDNA from hepatocytes. For this purpose, long-term suppression of HBV replication is the main endpoint of current therapeutic strategies, with the elimination of HBsAg as an optimal endpoint. Therefore, levels of serum ALT, HBV DNA, and HBsAg are important predictors of long-term prognosis.

The HBV genome is converted to cccDNA from relaxed circular (rc) DNA after its entry into the nucleus, using cellular DNA repair systems [7,8,9,10]. cccDNA transcribes virus-related mRNAs, of which pregenomic RNA (pgRNA) is a template for reverse transcription followed by plus-strand DNA synthesis to generate rcDNA, the viral genome. Since cccDNA lacks a replication origin, it cannot replicate by semi-conservative replication; thus, cccDNA is amplified by its conversion from rcDNA during *de novo* infection (*de novo* synthesis) or through recycling steps after intracellular rcDNA amplification (intracellular amplification) [11,12,13]. Once cccDNA is formed, it is stably pooled in the nucleus via intracellular recycling of the HBV genome [14]. Chronic HBV infection is caused by the persistence of cccDNA, which is transcriptionally competent for all HBV RNAs, in the nucleus of hepatocytes [15]. A recent study suggested that the *de novo* synthesis and intracellular amplification of cccDNA are differentially controlled by viral and/or cellular mechanisms [16].

DNA polymerase κ and λ, which are specifically involved in translesion synthesis and nucleotide excision DNA repair [17], and in meiotic recombination and DNA repair [18], support cccDNA formation in the *de novo* infection pathway [8]. On the other hand, DNA polymerase α has a role in repairing the minus strand in the conversion of rcDNA to cccDNA and supports the biosynthesis of cccDNA in the intracellular amplification cycle and the viral genome recycling step [19].

The dynamics of cccDNA during long-term culture was investigated by an in vitro infection assay system using NTCP-expressing HepG2 cells [14]. After infection, the cccDNA increased to 5 to 12 copies per cell over a period of 45 days. HBV cccDNA collapsed with a half-life of 40 days under treatment by entecavir (ETV), which inhibits intracellular recycling of the HBV genome by preventing rcDNA formation. When lamivudine was used in HBV-infected woodchucks and ducks, the half-lives were 33–50 and 35–57 days, respectively [20,21].

It is predicted that the mutation rates of essential genes such as polymerase are lower than those of non-essential genes [22,23,24]. However, the possibility of the emergence of drug-resistant mutants cannot be eliminated if the HBV DNA levels are not controlled by long-term administration of NRTIs. The final treatment goal of therapy for HBV-infected patients is to inhibit cccDNA amplification and finally eliminate cccDNA. Lately, numerous researchers have been trying to develop novel technologies such as genome editing for the elimination of cccDNA [25,26,27]. However, there are many issues to be solved before clinical trials begin: for instance, off-target effects, un-expected chromosomal DNA recombinations and cleavage of integrated HBV DNA. Unfortunately, we do not know the details of the biochemical processes of cccDNA formation and amplification, or the adverse effects of new technologies. A better understanding of the mechanisms of these events should be obtained before devoting extensive resources into the development of new therapies targeting cccDNA. At present, polymerase inhibitors are still the most feasible and powerful therapeutic agents for HBV therapy. In this review, we summarize recent studies of combination therapy using polymerase inhibitors and discuss the importance of non-nucleos(t)ide reverse transcriptase inhibitor (NNRTI) therapies for future treatment of HBV infection.

## 2. Potential of Polymerase Inhibitors

HBV polymerase is one of the most appropriate therapeutic targets because it has multiple and essential roles in the viral replication cycle (Figure 1). pgRNA is transcribed from cccDNA by cellular transcription machineries, and is transported to the cytoplasm, where it is packaged into a nucleocapsid by interacting with the polymerase through the “ε” (“epsilon” for encapsidation) packaging signal. Protein priming starts at position tyrosine 63 (Y63) of the HBV polymerase after RNA packaging, followed by RNA-dependent minus-strand DNA replication, namely, reverse transcription. While pgRNA is degraded by the RNase H activity of polymerase, rcDNA is produced by a DNA-dependent plus strand DNA synthesis, the details of which are described elsewhere [28,29]. Polymerase inhibitors strongly block rcDNA formation and, thereby, inhibit intracellular cccDNA amplification. To avoid further amplification of cccDNA after infection, development of novel NRTIs and NNRTIs is still essential. Until now, NRTIs are the only available direct-acting antivirals (DAAs) for HBV therapy; there are no NNRTIs against HBV polymerase, in contrast to the case for human immunodeficiency virus (HIV) reverse transcriptase (RT).

Deoxyribonucleotide triphosphates (dNTPs) are incorporated into nascent DNA by the enzymatic activity of HBV polymerase (Figure 2a), where NRTIs are converted to their triphosphate form in the cell, and are incorporated into nascent DNA by HBV polymerase similarly to natural dNTPs. These NRTIs compete with the natural dNTPs and stop DNA elongation due to their lack of a 3′-hydroxyl group, thereby terminating the incorporation of subsequent incoming nucleotides (Figure 2b) [30,31,32].

Currently, ETV, tenofovir disoproxil fumarate (TDF), and tenofovir alafenamide (TAF) are the first-line agents for the antiviral treatment of CHB in the clinical guidelines because of the high antiviral effect and low emergence rate of drug-resistant viruses [1,33,34]. Serum hepatitis B surface antigen (HBsAg) is used as a diagnostic marker of HBV infection and reflects the content of intrahepatic HBV cccDNA [35,36]. NRTIs efficiently suppress the viral DNA replication steps by inhibiting reverse transcription and DNA-dependent DNA replication, but they are not involved in the direct inhibition of cccDNA. However, recent studies have reported that NRTI treatment increased IFN-λ3 levels in both in vivo and in vitro experiments, inducing IFN-stimulated genes (ISGs), and then inhibited production of HBsAg in hepatoma cells, suggesting hitherto unknown, synergistic pharmacological effects of NRTIs [37,38].

In addition to NRTIs, pegylated interferon (peg-IFN) has been used in combination therapy, and although the cure rate (diagnosed by loss of HBsAg) of each alone is often very low, combined treatment with peg-IFN and NRTIs has shown a dramatic increase in the cure rate as described in Section 3.2. To overcome HBV infection by combination therapy, it is important not only to develop therapeutic drugs targeting various life cycles, but also to develop drugs targeting the polymerase itself, including NNRTIs. NNRTIs directly bind to the polymerase and cause a conformational change that disrupts the active site function, thereby leading to the loss of polymerization activity (Figure 2c). NNRTIs are one of the essential drugs in the combination therapies for HIV because of their strong antiviral activity, high specificity, and low toxicity.

Combinations of NRTIs and NNRTIs are widely used in HIV therapies. The mode of action of NNRTIs has been investigated by crystallography, structural analysis, and a docking model of HIV RT complexed with NNRTIs. NNRTIs bind to RT and inhibit polymerization through conformational changes of some residues of RT, and NNRTIs are not necessary for intracellular metabolism like NRTIs. Das et al. revealed the molecular mechanism of inhibition by nevirapine using structural analysis [39]. Binding of nevirapine opens the NNRTI-binding pocket, and the formation of this pocket causes the 3′ end of the DNA primer to move away from the position of the polymerase active site, which reduces nucleic acid interaction and strains the dNTP-binding pocket, resulting in inhibition of DNA synthesis. Unfortunately, efficient NNRTIs for HBV therapies have not yet been developed.

Structural information is always helpful for the design of drugs targeting specific molecules; unfortunately, the structural analysis of HBV polymerase has not progressed due to its highly insoluble protein character. This is one of the reasons why novel anti-HBV polymerase drugs have not been developed yet. Recent studies have reported the successful development of an advanced purification system for HBV polymerase using the partial domains having nucleotide- and template/primer-binding activity [40,41]. Highly advanced technologies for purification of polymerase will provide a new avenue for further development of novel NNRTIs.

## 3. Combination Therapy Is a Prerequisite for the Elimination of Virus

Long-term monotherapy with NRTI might induce the emergence of drug-resistant viruses. Therefore, combination therapy to inactivate viruses from multiple angles is an attractive strategy as a treatment against viral infections because of its additive and possibly synergistic inhibitory effects against even drug-resistant mutants.

### 3.1. Combination Therapies against Human Immunodeficiency Virus (HIV) and Hepatitis C Virus (HCV)

Highly active antiretroviral therapy (HAART) has significantly reduced the incidence of HIV infection and the mortality of HIV-infected patients [42]. HAART targets multiple viral replication cycles using a combination of drugs including NRTI, NNRTI, protease inhibitor (PI), and/or integrase inhibitor types [43,44]. HIV reverse transcriptase (RT) has an essential role for converting ssRNA to ssDNA, i.e., reverse transcription, and is one of the important targets of HAART [45]. NNRTIs bind to an allosteric region to form a complex with HIV RT, changing its conformation and hence impeding its function [46,47,48]. A recent study demonstrated that a new NNRTI and NRTI combination therapy showed a strong synergistic inhibitory effect even on mutant strains that are resistant to an NRTI or NNRTI when used alone [49].

The development of new combination therapies has led to a dramatic therapeutic outcome for hepatitis C virus (HCV) infection as well. Drugs targeting the viral polymerase significantly increased the cure rate [50,51,52]. Sofosbuvir, which is a potent nucleotide analog against HCV polymerase NS5B, and ledipasvir, which is a polymerase associate factor NS5A inhibitor, have been used in combination for the treatment of chronic HCV since 2014, and dramatic therapeutic effects have been shown [53,54,55]. Sofosbuvir suppresses ledipasvir-resistant mutants with the combination of sofosbuvir and ledipasvir in an HCV replicon cell line [56]. These studies indicated that the combination of NRTIs and NNRTIs in the treatment of either HIV or HCV can provide a drastic inhibitory effect.

### 3.2. Combination with Interferon (IFN) and Nucleos(t)ide RT Inhibitors (NRTIs) for Hepatitis B Virus (HBV) Therapy

In case of HBV, it was reported that combination therapy using interferon and NRTIs effectively increased the cure rate. Hagiwara et al. reported the effectiveness of a simultaneous combination of ETV and peg-IFNα. CHB patients treated with ETV and peg-IFNα for 48 weeks showed reduced HBsAg levels, regardless of their HBeAg-positivity; the loss rate of HBsAg was 3.8% at 1 year, 8.4% at 3 years, and 15% at 5 years after treatment [57,58]. Another study showed that after 48 weeks of treatment with both TDF and IFN, 13% of the HBV patients (both HBeAg-positive and -negative) lost HBsAg, whereas only 3% of the patients treated with only Peg-IFNα lost HBsAg [59].

Sequential “switch to combination”, which starts with one therapy followed by another therapy, has been shown to be more effective than simultaneous-combination treatment. Several reports have shown that loss rate of HBsAg increased from 32 to 36% in CHB patients by the sequential administration of a 1- or 2-year course of NRTI (predominantly ETV) followed by interferon for 48 or 60 weeks. On the other hand, only 0 or 4.3% of patients showed HBsAg loss by administration of an NRTI alone [60,61].

The fundamental strategy of HBV therapy is long-term NRTI treatment, because potent NRTI treatment results in excellent viral suppression (more than 95% at 5 years) [1]. Sequential “add-on” combination therapy, in which interferon is added to ongoing NRTI treatment, has also been studied. Campenhout et al. reported that the peg-IFN add-on treatment resulted in twice as many patients achieving a decrease in HBsAg levels of more than 1 log compared to ETV monotherapy (add-on therapy, 59%; ETV monotherapy, 28%), and the loss of HBsAg was observed in 2.1% patients with add-on treatment [62]. Bourlière et al. reported that at 144 weeks, 10% of patients with add-on combination treatment had lost HBsAg compared to 4% of patients with the NRTI monotherapy [63].

As described above, many studies have reported that simultaneous, sequential, or add-on combination therapy using ETV or TDF with peg-IFNα provides more effective therapeutic results. It has been noted that the choice of therapy should be carefully determined for each patient, because adverse effects to interferon vary among individuals.

### 3.3. Combination Treatment with NRTIs and Other Agents

Several in vitro and in vivo studies have suggested the potent inhibitory effects of combination treatment of NRTI with chemical agents. Zhu et al. reported that the combination of tenofovir with either emtricitabine (FTC), lamivudine (3TC), ETV, telbivudine (LdT) or adefovir (AFV) showed additive or synergistic inhibition effects, both in vitro and in a mouse model [64]. Zhen et al. reported that the combination of an NRTI and an anti-PD-1 antibody resulted in the inhibition of viral gene expression and improvement of the survival of HBV transgenic mice [65]. Additionally, the combination treatment enhanced the production of interferons from T cells, and increased the expression of Th1-related immunostimulatory genes, resulting in reduction of the transcription of regulatory and inhibitory immune genes. These results suggested that a combination targeting HBV and blocking the PD-1 immune checkpoint should have a strong synergistic effect. Moreover, synergistic anti-HBV activity and anti-HBV replication activity has been shown by the combination of LdT with saikosaponin c isolated from the herbal drug, Radix Bupleuri [66]. In addition, saikosaponin c inhibits pgRNA synthesis by stimulating IL-6 expression, resulting in the attenuation of HNF1α and HNF4α expression [67]. ABI-H0731, an HBV core protein inhibitor, has exhibited significant antiviral activity in phase 1b clinical trial in CHB patients [68]. ABI-H0731 directly targets HBV core protein, preventing HBV pgRNA encapsidation, thereby inhibiting HBV DNA replication. The combination of ABI-H0731 with ETV shows an additive to moderately synergistic effect.

## 4. Discussion and Perspectives

Silencing or depleting the cccDNA pool in infected hepatocytes is the goal for new approaches of the treatment. Direct targeting of cccDNA is a strong therapeutic strategy, and polymerase inhibitors play an essential role for preventing further cccDNA accumulation via prevention of the intracellular recycling steps and of viral DNA replication itself. The active development of new therapeutic options to inhibit various life-cycle steps, including cccDNA formation, is ongoing.

Nevertheless, HBV polymerase is absolutely the most effective therapeutic target to drastically reduce viral replication. Although genome editing to directly eliminate cccDNA would provide an innovative therapy, problems such as off-target effects, unexpected chromosomal DNA recombination, and cleavage of integrated HBV DNA must be solved, as mentioned above.

NNRTIs against HIV are available for the current therapies, but in the case of HBV, there have been no available NNRTIs until now. The development of NNRTIs against HBV based on the nature of the HBV polymerase is one of the most important issues to be addressed in order to achieve dramatic therapeutic effects with a combination therapy and to increase treatment options for resistant viruses.

Failure to control viral replication during prolonged monotherapy increases the risk of the emergence of drug-resistant viruses. The emergence of drug-resistant mutant strains of HBV occurs frequently due to the use of monotherapy with antivirals that are less potent and have a lower genetic barrier to resistance [69]. Previously, long-term monotherapy with ADV and 3TC resulted in a high incidence of drug-resistant mutations, but with the advent of ETV, TDF and TAF, the incidence of resistant mutations has decreased [1]. In addition, recent HIV study has reported that NRTIs in combination with NNRTIs also have a synergistic inhibitory effect on NRTI-resistant mutants [49], suggesting that combination therapy may be effective in inhibiting the growth and emergence of drug-resistant strains. Therefore, if rapid and sustainable control of viral replication can be achieved by combination therapy using multiple targeted therapeutic agents including polymerase, the risk of the emergence of resistant viruses can be reduced. For this purpose, the development of combination strategies targeting different viral life stages including viral genome replication is needed to improve the cure rate of chronic hepatitis B.

Taking into consideration the successful treatment of HCV and HIV by combination therapy and the strict control of HBV DNA levels by NRTIs, combination therapy of NRTIs with interferons or other inhibitors can be expected to provide improved therapeutic responses in HBV infection.

## Figures and Tables

**Figure 1 viruses-13-01691-f001:**
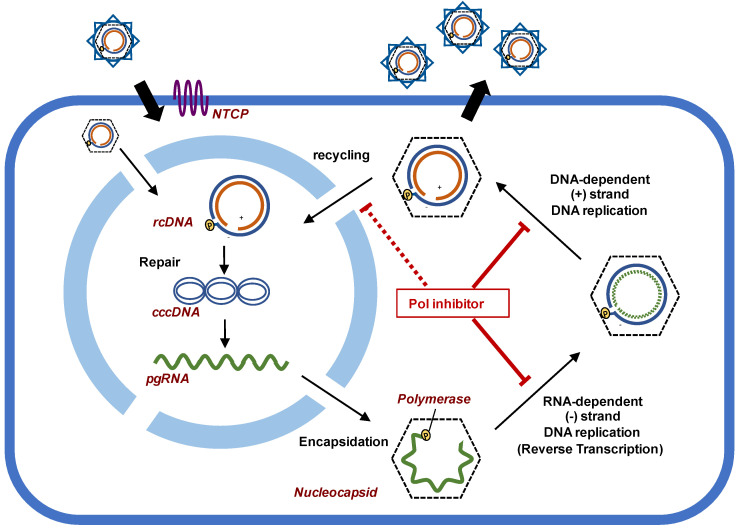
The lifecycle of hepatitis B virus (HBV) focusing on viral genome replication. HBV polymerase has several crucial roles in viral replication. Polymerase inhibitors block the reverse transcription pathway, namely RNA-dependent minus-strand DNA synthesis; they also block DNA-dependent plus-strand DNA synthesis, thereby suppressing the recycling step for cccDNA amplification.

**Figure 2 viruses-13-01691-f002:**
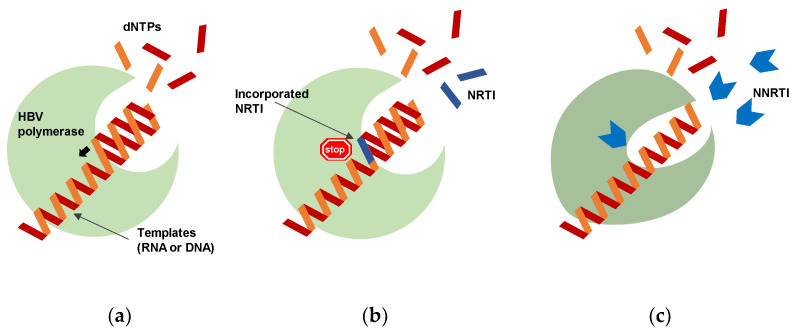
Different roles between nucleos(t)ide RT inhibitors (NRTIs) and non-nucleos(t)ide RT inhibitors (NNRTIs). (**a**) Deoxyribonucleotide triphosphates (dNTPs) are incorporated into nascent DNA during reverse transcription (minus-strand DNA synthesis) and plus-strand DNA synthesis. (**b**) NRTIs block DNA synthesis via chain termination by incorporating themselves into the nascent DNA. (**c**) NNRTIs block DNA synthesis via direct binding to the polymerase and causing enzyme conformational changes that disrupt active-site function, leading to impairment of polymerization activity.

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
