# Peer review of "Chronic Hepatitis B Treatment Strategies Using Polymerase Inhibitor-Based Combination Therapy"

_viruses, 2021, doi:10.3390/v13091691_

Round 1
Reviewer 1 Report
This an informative article, especially for clinicals. Though the article is well written with most updated information, following concerns should be addressed by the authors.
Title should be amended to "Chronic hepatitis B treatment strategies using polymerase inhibitor-based combination therapy" or "Treatment strategies using polymerase inhibitor-based combination therapy for Chronic hepatitis B virus infection".
Line 10: Please correct as ".....difficulty of eliminating covalently...."
Line 10-13: Rephrase the sentence "Recently, many researchers have been trying to......capsid assembly, egress, and so on".
Line 29-30: Define "ALT" and "HBV antigens"
Line 36: Define CHB at first use in Line 30.
Line 42: Mention with clarity- "HBV polymerase gene mutants or variants".
Line 69-71: Rephrase "DNA polymerase κ, which..... in the de novo infection pathway [8]".
Line 101-103: Repeated statement "After infection, the viral rcDNA... transformed into cccDNA by the cellular repair system [28–31].
Line 181: Subsection "3.1. Combination therapies against HIV and HCV" can be deleted.
Author Response
Please see the attachment PDF file.

Reviewer 2 Report
This is a generally balanced review on NRTIs against HBV infection. It is important to include drug resistant mutations of HBV against NRTIs. In addition, the manuscript will be better if the authors can discuss the possible impact by the new antivirals under clinical trial on current NTRI-based treatment.
Author Response
Please see the attachment PDF file.

Round 2
Reviewer 2 Report
The authors have addressed my concerns. The manuscript is acceptable for publication.